# Uptake of osteoblast-derived extracellular vesicles promotes the differentiation of osteoclasts in the zebrafish scale

Jingjing Kobayashi-Sun[1], Shiori Yamamori[1], Mao Kondo[1], Junpei Kuroda[2], Mika Ikegame[3], Nobuo Suzuki[4], Kei-ichiro Kitamura[5], Atsuhiko Hattori[6], Masaaki Yamaguchi[7] & Isao Kobayashi[7✉]

Differentiation of osteoclasts (OCs) from hematopoietic cells requires cellular interaction with osteoblasts (OBs). Due to the difficulty of live-imaging in the bone, however, the cellular and molecular mechanisms underlying intercellular communication involved in OC differentiation are still elusive. Here, we develop a fracture healing model using the scale of *trap: GFP*; *osterix:mCherry* transgenic zebrafish to visualize the interaction between OCs and OBs. Transplantation assays followed by flow cytometric analysis reveal that most *trap:GFP*high OCs in the fractured scale are detected in the *osterix:mCherry*+ fraction because of uptake of OB-derived extracellular vesicles (EVs). In vivo live-imaging shows that immature OCs actively interact with *osterix:mCherry*+ OBs and engulf EVs prior to convergence at the fracture site. In vitro cell culture assays show that OB-derived EVs promote OC differentiation via Rankl signaling. Collectively, these data suggest that EV-mediated intercellular communication with OBs plays an important role in the differentiation of OCs in bone tissue.

[1] Division of Life Sciences, Graduate School of Natural Science and Technology, Kanazawa University, Kanazawa, Ishikawa 920-1192, Japan. [2] Graduate School of Frontier Biosciences, Osaka University, Suita, Osaka 565-0871, Japan. [3] Department of Oral Morphology, Graduate School of Medicine, Dentistry and Pharmaceutical Sciences, Okayama University, Okayama, Okayama 700-8525, Japan. [4] Noto Marine Laboratory, Institute of Nature and Environmental Technology, Division of Marine Environmental Studies, Kanazawa University, Noto-cho, Ishikawa 927-0553, Japan. [5] Department of Clinical Laboratory Science, Division of Health Sciences, Graduate School of Medical Science, Kanazawa University, Kanazawa, Ishikawa 920-0942, Japan. [6] Department of Biology, College of Liberal Arts and Sciences, Tokyo Medical and Dental University, Ichikawa, Chiba 272-0827, Japan. [7] Faculty of Biological Science and Technology, Institute of Science and Engineering, Kanazawa University, Kanazawa, Ishikawa 920-1192, Japan. ✉email: ikobayashi@se.kanazawa-u.ac.jp

Over the past few decades, the zebrafish and medaka have emerged as attractive genetic models in a variety of research fields due to their many unique advantages, including external development, large offspring numbers, transparency of embryos, short generation time, and availability of transgenic and mutant lines. In the field of bone research, some transgenic lines have been generated to visualize osteoclasts (OCs) and osteoblasts (OBs) in zebrafish and medaka[1,2]. The zebrafish scale is a thin membranous bone embedded in the skin that consists of OBs, OCs, and bone matrix. Although the structure of zebrafish scales is rather simple compared with mammalian bones[3–5], various in vivo and in vitro studies have demonstrated that OCs and OBs in teleost scales respond to hormones and other substances as one would predict from observation of the mammalian bone[6–12], indicating that fundamental cellular and molecular programs that regulate OCs and OBs are highly conserved amongst vertebrates. In addition, this "surface" bone tissue enables in vivo live-imaging of OCs and OBs using fluorescent transgenic lines.

OCs differentiate from hematopoietic stem cells and fuse to become mature multinucleated OCs, which can resorb bone via secretion of hydrochloric acid[13,14]. In contrast, OBs arise from mesenchymal stem cells and can produce bone matrix in response to a wide variety of growth factors[15]. OBs also regulate the formation and activity of OCs via signaling molecules[16], highlighting the importance of cell–cell communication in osteoclastogenesis.

There are two well-documented modes of intercellular communication in OC differentiation. The first mode is through direct contact between OBs and OC precursors, allowing membrane-bound ligands and receptors to interact and initiate intercellular signaling (e.g. receptor activator of nuclear factor kappa B ligand (RANKL)-RANK signaling). The second mode is dependent on diffusible paracrine factors, including cytokines secreted by OBs and acting on OC precursors via diffusion (e.g. macrophage-colony stimulating factor (M-CSF))[17–19]. Recent studies indicate that OB-derived extracellular vesicles (EVs) can be considered a third mode of communication in OC differentiation. Two groups demonstrated that EVs shed from OBs contain RANKL proteins and can transfer signals to OC precursors through the cell-surface receptor RANK, leading to the formation of OCs[20,21]. The molecular composition of EVs is thought to be strictly regulated in the releasing cell by the external stimuli[22,23]. Multiple studies have consistently demonstrated that EVs transfer proteins, lipids, and RNAs between various cell types, thus mediating intercellular communication[24,25]. It is also reported that OC-derived EVs promote OB differentiation or inhibit osteoclastogenesis[26,27]. Due to the difficulty of live-imaging in the bone, however, it is still unknown if immature OCs actually obtain OB-derived EVs in vivo to promote their differentiation.

In the present study, we generated a double transgenic zebrafish, trap:GFP; osterix:mCherry, which labels OCs and OBs with GFP and mCherry, respectively. Combined with intubation anesthesia, it is possible to visualize and trace the dynamics of OCs and OBs during the fracture-healing process in the scale. Taking advantage of this system, we uncovered that immature OCs engulf OB-derived EVs under fracture stress, leading to the differentiation of OCs.

## Results

### Convergence and fusion of OCs in the fractured scale.
To visualize OCs and OBs in the zebrafish scale, we generated a double-transgenic zebrafish, trap:GFP; osterix:mCherry, which expresses GFP and mCherry under the control of the OC-specific trap (tartrate-resistant acid phosphatase, also known as acid phosphatase 5a, tartrate resistant (acp5a)) and OB-specific osterix (also known as sp7 transcription factor) enhancer, respectively. Confocal imaging of extracted scales revealed that osterix:mCherry+ cells were distributed throughout the scale including the epidermis and dermis area under physiological conditions, while osterix:mCherry[bright] cells were limited at the edge region of the scale. In contrast, only a few, mostly small and round trap:GFP+ cells were observed (Fig. 1a). In order to induce fracture stress in the scale, the epidermis area of the scale was cut with fine scissors and confocal imaging was performed at 2 days post-fracture (dpf). We found that the fracture site was surrounded by many trap:GFP[bright] cells having more than 10 nuclei (Fig. 1b), indicating that multinucleated OCs are formed under fracture stress.

To further examine fracture healing in the scale, in vivo time-course imaging was performed using an intubation anesthesia system (Supplementary Fig. 1). As shown in Fig. 1c, some small round trap:GFP+ cells converged in the vicinity of the fracture site at 1 dpf. Interestingly, many trap:GFP+ cells were detected around the edge region near the fracture site where osterix:mCherry[bright] cells are abundantly observed. These trap:GFP+ cells appeared to closely interact with osterix:mCherry[bright] cells. At 2 dpf, trap:GFP+ cells in the edge region decreased, but a large number of trap:GFP[bright] cells appeared and covered the fracture site. These observations suggest that immature OCs interact with OBs mainly in the edge region of the scale prior to reaching the fracture site. After 4 dpf, the number of trap:GFP+ cells began to decrease, and the fracture site was covered with osterix:mCherry+ cells instead, leading to bone formation at the fracture site by OBs. Time-lapse imaging of the fractured scale in trap:GFP animals revealed that actively migrating trap:GFP+ cells fused to generate multinucleated OCs around the fracture site at 1 dpf. These fused cells were larger in size and brighter in GFP expression compared to unfused trap:GFP+ cells (Supplementary Movies 1 and 2). These data indicate that trap:GFP and osterix:mCherry finely label OCs and OBs, respectively, and that the fractured scale model is useful to investigate the differentiation process of OCs in vivo.

### OCs possess mCherry+ particles in the cytoplasm.
To isolate and characterize OCs and OBs in the fractured scale, cells were collected from scales in trap:GFP; osterix:mCherry double-transgenic animals and analyzed by flow cytometry (FCM). We detected trap:GFP+ cells and osterix:mCherry+ cells in both intact and fractured scales at 1 dpf. Unexpectedly, however, most trap:GFP+ cells were detected in the osterix:mCherry+ fraction in the scale (Fig. 2a). The percentage of trap:GFP[low] osterix:mCherry+ ("GFP[low]") and trap:GFP[high] ("GFP[high]") cells significantly increased in fractured scales compared with intact scales (Supplementary Fig. 2a). Within the trap:GFP[high] fraction, 75.1 ± 3.8% of cells were detected in the mCherry+ fraction ($n = 6$, ±s.d.) (Supplementary Fig. 2b). We also examined the absolute number of trap:GFP− osterix:mCherry+ ("mCh+"), GFP[low], and GFP[high] cells in an intact or fractured scale. The number of GFP[low] and GFP[high] cells was approximately 2.1 and 3.7 times higher in the fractured scale compared with the intact scale, respectively, whereas that of mCh+ cells was unchanged (Fig. 2b).

Cells within the mCh+, GFP[low], and GFP[high] fraction at 1 dpf were separately sorted, plated on a fibronectin-coated glass-bottom dish, and analyzed by confocal microscopy. Cells within the mCh+ fraction contained at least three different types of OBs: small and round cells (type-1), small and spindle-shaped cells (type-2), and large, highly spread cells (type-3). Type-1 mCh+ cells were most frequently observed (~60%) (Fig. 2c). Cells within the GFP[low] fraction appeared mostly small and round and were

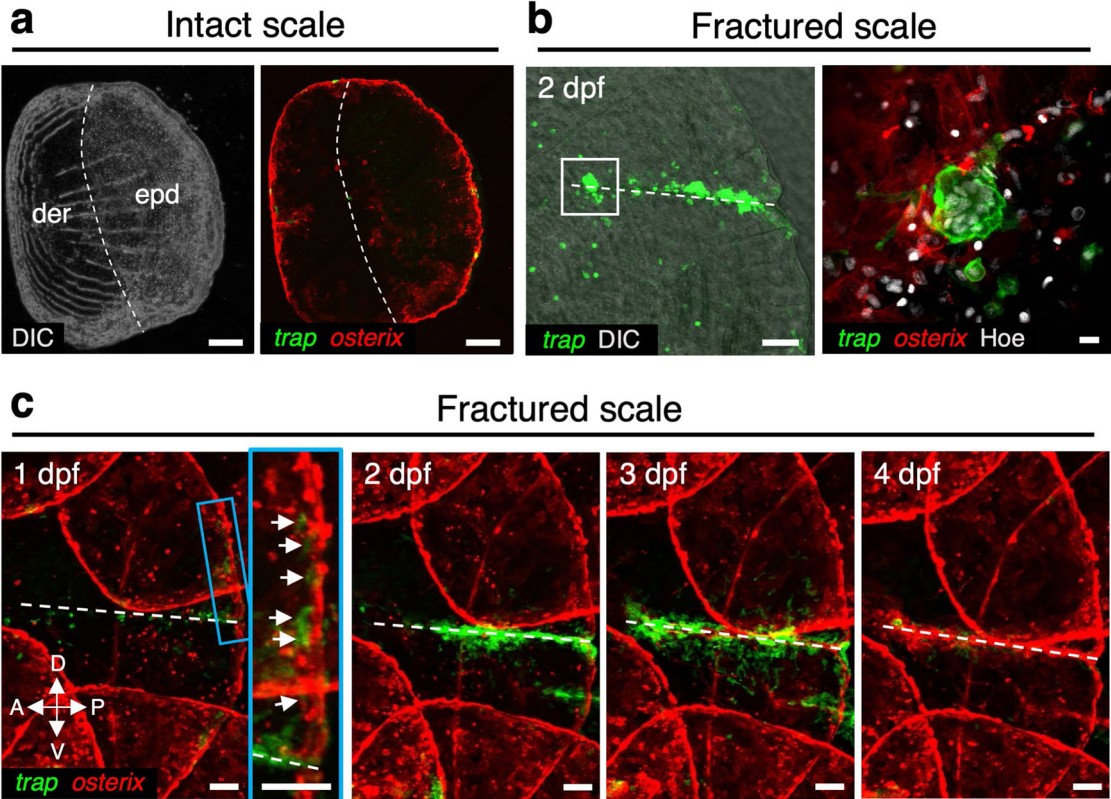

**Fig. 1 _trap:GFP_[+] cells converged at the fracture site in the zebrafish scale. a**, **b** Representative images of an intact **a** or fractured scale **b** in _trap:GFP; osterix:mCherry_ double-transgenic zebrafish. Dotted lines in **a** show a boundary of the dermis (der) and epidermis (epd) area. The right panel in **b** shows a high magnification view of the white boxed area in the left panel. **c** Representative time-course changes of a fractured scale. The inset in the left panel shows a high magnification view of the blue boxed area. Images are orientated with the dorsal side to the top and anterior side to the left. Arrows indicate _trap:GFP_[+] cells observed in the edge area of the fractured scale. Dotted lines in **b** and **c** show the fracture site. DIC differential interference contrast. Hoe Hoechst 33342; dpf days post-fracture; bars, 200 μm **a**; 10 μm (right panel in **b**); 100 μm (left panel in **b**, **c**). Experiments were performed twice with three biological replicates in each group **a**–**c**.

mononucleated (type-1). Some GFP[low] cells did not express GFP diffusely, but contained GFP[+] fragments in the cytoplasm, appearing to be phagocyte-like cells (type-2) or mCh[+] OBs (type-3) (Fig. 2d). In contrast, cells within the GFP[high] fraction contained different types of OCs. Some cells were small and round with a single nucleus (type-1), while others exhibited an ameba-like morphology with many pseudopodia and one or two nuclei (type-2). A few GFP[high] cells had three to four nuclei and were relatively larger in size (type-3), suggesting that various stages of OCs may exist in the GFP[high] fraction (Fig. 2e). We also found that type-2 OCs very actively migrated in vitro in contrast to type-1 and type-3 OCs, which exhibited low motility (Supplementary Movies 3 and 4). Interestingly, most _trap:GFP_[+] cells possessed mCherry[+] particles in the cytoplasm (Fig. 2d, e). Electron microscopic analysis revealed that GFP[high] cells had small protrusions, irregular-shaped nuclei, abundant mitochondria, and compact Golgi apparatus located close to the nucleus (Fig. 2f), which are representative morphological features of OCs[28]. We also observed various types of vesicles, including secondary lysosomes, early endosomes, and multi-vesicular bodies, in the cytoplasm of GFP[high] cells (Fig. 2f).

**Transplantation confirms uptake of OB-derived EVs in OCs.** Having shown that most _trap:GFP_[+] cells possessed mCherry[+] particles, we hypothesized that immature OCs engulf OB-derived EVs to become mature OCs under fracture stress. Since OCs have been shown to originate from hematopoietic stem cells in

mammals[29], we performed two different hematopoietic cell transplantation assays in order to examine if donor-derived OCs can obtain OB-derived particles in the recipient scale. First, kidney marrow cells (KMCs), which contain a variety of hematopoietic cells including hematopoietic stem cells[30,31], were collected from _trap:GFP; osterix:mCherry_ double-transgenic animals and were transplanted into wild type recipients irradiated with sublethal dose of X-ray. At 20 weeks post-transplantation, fractured scales in recipients were analyzed by FCM or confocal microscopy (Fig. 3a, "Transplantation-1"). We detected only _trap: GFP_ single-positive cells, but not _osterix:mCherry_-expressing cells, in the scale of recipients (Fig. 3b, c; $n = 3$), indicating that _trap: GFP_[+] OCs originate from hematopoietic stem cells in zebrafish, as has been shown in mammals[29]. We next transplanted KMCs from _trap:GFP_ single-transgenic animals into irradiated _osterix: mCherry_ single-transgenic recipients (Fig. 3a, "Transplantation-2"). In this experiment, we detected GFP and mCherry double-positive cells in the fractured scale of _osterix:mCherry_ recipients. The mean percentage of mCherry[+] cells within the _trap:GFP_[high] fraction was approximately $48.1 \pm 13.5\%$ ($n = 4$, ±s.d.) in recipient scales (Fig. 3b).

Supporting these observations, live-imaging analysis of the fractured scale in a _trap:GFP; osterix:mCherry_ double-transgenic animal revealed that a _trap:GFP_[+] OC actively interacted with _osterix:mCherry_[+] OBs and obtained mCherry[+] particles prior to reaching the fracture site (Supplementary Movie 5). Interestingly, we observed that a _trap:GFP_[+] cell extended the protrusion toward

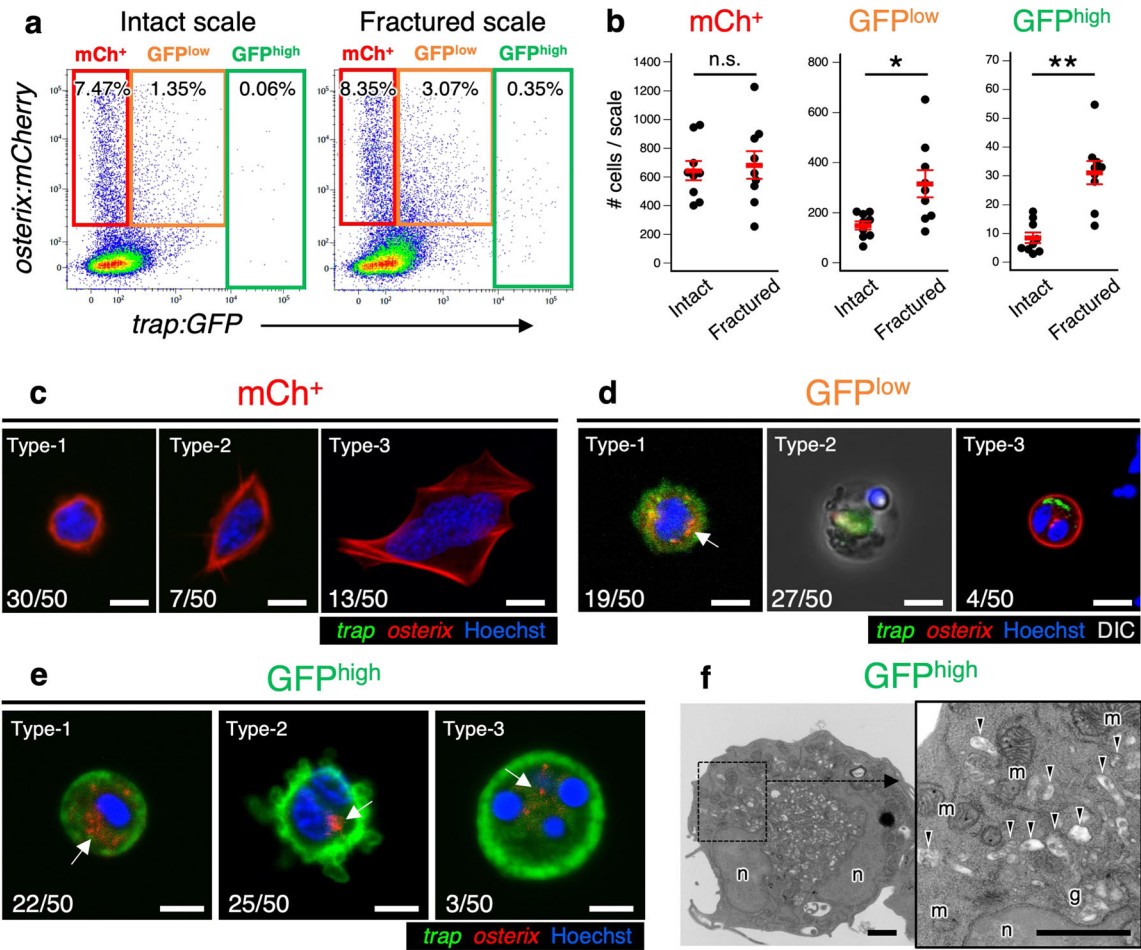

**Fig. 2 OCs contain *osterix:mCherry*+ particles. a** Representative results of flow cytometric analysis of cells from intact (left panel) or fractured scales at 1 day post-fracture (dpf) (right panel). Red, orange, and green gate show *trap:GFP*− *osterix:mCherry*+ ("mCh+"), *trap:GFP*low *osterix:mCherry*+ ("GFPlow"), and *trap:GFP*high ("GFPhigh") cells, respectively. **b** Absolute number of mCh+, GFPlow, and GFPhigh cells in an intact or fractured scale at 1 dpf. Error bars, s.e.m. (*n* = 9 for each group); n.s., no significance; \**p* < 0.05; \*\**p* < 0.001 by Student's *t*-test. **c–e** Representative fluorescent images of mCh+ **c**, GFPlow **d**, and GFPhigh cells **e**. Arrows indicate an *osterix:mCherry*+ particle observed in the cytoplasm. Numbers in bottom left of panels indicate the number of cells showing the displayed morphology over the total number of analyzed cells. DIC differential interference contrast. **f** Representative electron microscopic images of a GFPhigh cell. Arrowheads show vesicles, which include secondary lysosomes, early endosomes, and multi-vesicular bodies. n nucleus; m mitochondrion; g Golgi apparatus; bars, 5 μm **c–e**; 1 μm **f**. Experiments were performed twice with nine biological replicates **a**, **b** and two biological replicates **c–f** in each group.

mCherry+ particles and engulfed them into the cytoplasm (Fig. 4, Supplementary Movie 6). Collectively, these data suggest that *trap:GFP*+ OCs obtain OB-derived EVs in the fractured scale.

**OB-derived EVs contain abundant signaling molecules.** To isolate and characterize OB-derived EVs in the scale, we next attempted to isolate OB-derived EVs in fractured scales by FCM using Hoechst 33342 (Hoe), a fluorescent dye that stains the DNA in living cells. Cells dissected from fractured scales at 1 dpf were stained with Hoe and analyzed by FCM. We detected not only an mCh+ Hoehigh fraction but also an mCh+ Hoelow fraction, in which non-nucleated *osterix:mCherry*+ particles with very low forward scatter (FSC) intensity were identified (Fig. 5a). Ultrastructure of isolated mCh+ Hoelow particles showed that a part of particles, especially large particles, contained some cell compartments, including mitochondria and vesicles, while small particles showed vesicular nature without cellular organelles (Fig. 5b). Negative staining followed by electron microscopic analysis revealed that most mCh+ Hoelow particles were 0.6–1.5 μm in diameter, whereas a minority of particles were more than 2

μm (Fig. 5c, d). We found that the number of mCh+ Hoelow particles was approximately three times higher in the fractured scale than the intact scale, suggesting that OB-derived EVs are released in response to fracture stress (Fig. 5e).

To further characterize OCs, OBs, and OB-derived EVs, we performed RNA-seq analysis on four different populations in the fractured scale at 1 dpf: GFP− mCh+ Hoehigh ("mCh+" fraction), GFPlow mCh+ Hoehigh ("GFPlow" fraction), GFPhigh Hoehigh ("GFPhigh" fraction), and mCh+ Hoelow ("EV" fraction) (Fig. 6a). Principal component analysis (PCA) showed that GFPlow and GFPhigh were very closely associated, whereas mCh+ and EV were far from each other (Fig. 6b). We found that OC-related genes, such as *nfatc1* (*nuclear factor of activated T cells 1*), *ctsk* (*cathepsin K*), *mmp9* (*matrix metallopeptidase 9*), *csk* (*C-terminal Src kinase*), *itgb3b* (*integrin beta 3b*), and *atp6v* (*ATPase, H + transporting V*) family genes, were enriched in the GFPhigh fraction (Fig. 6c). In contrast, OB-related genes, such as *alpl* (*alkaline phosphatase, biomineralization associated*), *col1a1a* (*collagen, type I, alpha 1a*), *runx2b* (*RUNX family transcription factor 2b*), *wnt10b* (*wingless-type MMTV integration site family, member 10b*), *efnb2a* (*ephrin-B2a*), *csf1a* (*colony stimulating*

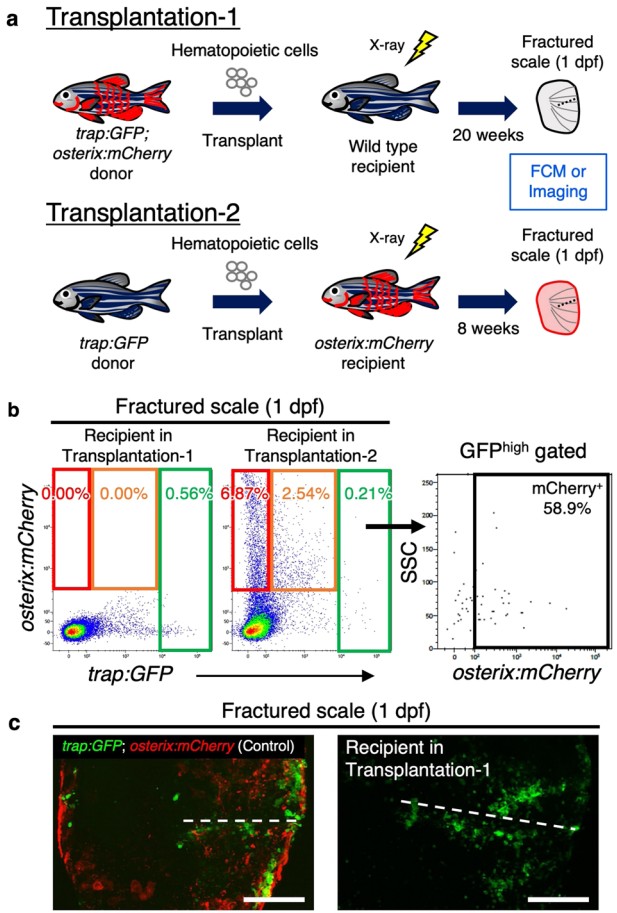

**Fig. 3 Transplantation assays confirm uptake of OB-derived EVs in OCs.** **a** Schematic diagram of transplantation assays. Hematopoietic cells from *trap:GFP*; *osterix:mCherry* double-transgenic zebrafish kidney were transplanted into wild type recipients irradiated with sublethal dose of X-ray (Transplantation-1, $n = 3$). Hematopoietic cells from *trap:GFP* single-transgenic zebrafish kidney were transplanted into *osterix:mCherry* single-transgenic recipients (Transplantation-2, $n = 4$). After 20 or 8 weeks post-transplantation, cells in scales at 1 day post-fracture (dpf) were analyzed by flow cytometry (FCM) and/or confocal microscopy. **b** Representative results of FCM analysis in fractured scales from a recipient in Transplantation-1 (left) and recipient in Transplantation-2 (middle). Red, orange, and green gate show *trap:GFP⁻ osterix:mCherry⁺* ("mCh⁺"), *trap: GFP^low osterix:mCherry⁺* ("GFP^low"), and *trap:GFP^high* ("GFP^high") cells, respectively. GFP^high cells in a recipient of Transplantation-2 are displayed in an *osterix:mCherry* vs. side scatter (SSC) dot plot (right panel). **c** Representative images of a fractured scale from a *trap:GFP*; *osterix:mCherry* double-transgenic animal (left) and recipient in Transplantation-1 (right). Dotted lines indicate the fracture site. Both images showed merged channels of GFP and mCherry. Bars, 100 μm. Experiments were performed twice with three or four biological replicates in each group.

*factor 1a*), and *tgfb1a* (*transforming growth factor, beta 1a*), were highly expressed in the mCh⁺ fraction (Fig. 6d).

To further determine the characteristics of each fraction, gene ontology enrichment analysis was performed using significantly up-regulated genes in each fraction. Genes involved in "proton transmembrane transport" and "ATP metabolic process" were predominantly expressed in the GFP^high fraction, highlighting that cells in the GFP^high fraction possess the representative molecular signature of OCs. In contrast, genes involved in "leukocyte activation" and "immune system process" were up-regulated in the GFP^low fraction (Fig. 6e). Based on morphological analysis and expression data, cells within the GFP^low fraction appear to contain mainly monocytes/macrophages that are induced by fracture stress. In contrast, genes involved in "extracellular matrix organization" and "response to wounding" were highly expressed in the mCh⁺ fraction. Interestingly, genes involved in "vesicle-mediated transport", "intracellular signal transduction", and "phagocytosis" were enriched in the EV fraction (Fig. 6e), suggesting that OB-derived EVs contain abundant signaling molecules.

Quantitative PCR analysis also showed that *trap*, *nfatc1*, and *ctsk* were highly expressed in the GFP^high fraction, whereas *osterix*, *alpl*, *col1a1a*, and *osteocalcin* (also known as *bone gamma-carboxyglutamate protein* (*bglap*)) were enriched in the mCh⁺ fraction (Supplementary Fig. 3). It is known in mammals that OC precursors express a cell-surface RANK, while OBs express its ligand RANKL, on the surface of their plasma membrane. This RANKL–RANK signaling activates a variety of downstream signaling pathways required for OC differentiation[32]. The expression of both *rank* and *rankl* was predominantly detected in both the GFP^low and EV fractions in the fractured scale (Supplementary Fig. 3). These data raised the possibility that OB-derived EVs can potentially induce the differentiation of OCs.

**OB-derived EVs promote OC differentiation via Rankl signals.** In general, EVs are classified into three types according to their sizes and origins: exosomes, microvesicles (MVs), and apoptotic bodies (ABs)[33]. Exosomes and MVs are shed from a variety of cell types and have been implicated in cell–cell communication[34]. ABs are formed during the apoptotic process and are then engulfed by macrophages and other immune cells[35,36]. Since exosomes are recognized as very tiny vesicles (<100 nm)[37,38], it is likely that the OB-derived EVs isolated by FCM and visualized by confocal microscopy are mainly MVs and/or ABs rather than exosomes. To distinguish MVs and ABs, we further performed cell staining with Sytox Red and Annexin-V, which distinguish MVs and ABs as well as live, pre-apoptotic, and apoptotic cells[39]. FCM analysis showed that mCh⁺ Hoe^high cells, which mainly contain OBs, were subdivided into three fractions, Sytox Red^low Annexin V^low (live cell fraction), Sytox Red^low Annexin V^high (pre-apoptotic cell fraction), and Sytox Red^high Annexin V^high (apoptotic cell fraction). We found that 79.2 ± 2.4% of mCh⁺ Hoe^high cells were detected in the live cell fraction, and 7.1 ± 1.3% and 11.2 ± 1.0% were in the pre-apoptotic and apoptotic cell fraction, respectively ($n = 4$, ±s.d.). Interestingly, 65.4 ± 5.0% of mCh⁺ Hoe^low EVs were detected in the Sytox Red^low Annexin-V^low "MV" fraction, while 32.2 ± 5.9% were in Sytox Red^low Annexin-V^high "AB" fraction ($n = 4$, ±s.d.) (Fig. 7a, b), indicating that mCh⁺ Hoe^low EVs contain both MVs and ABs.

To determine if OB-derived EVs are involved in the differentiation of OCs, 60,000 KMCs from *trap:GFP* animals were co-cultured with 2000 mCh⁺ Hoe^high Sytox Red^low Annexin-V^low "OBs", mCh⁺ Hoe^low Sytox Red^low Annexin-V^low "MVs", or mCh⁺ Hoe^low Sytox Red^low Annexin-V^high "ABs" on fibronectin-coated plates. After 2 days of culture, the number of *trap:GFP⁺* cells was counted in each well (Fig. 7c). We found that the number of GFP⁺ cells significantly increased in KMCs co-cultured with MVs or OBs when compared to non-co-cultured controls. Surprisingly, the number of GFP⁺ cells also significantly increased in KMCs co-cultured with ABs (Fig. 7d). Importantly, most *trap:GFP⁺* cells co-cultured with OBs, MVs, or ABs contained mCherry⁺ EVs in the cytoplasm (Fig. 7e), similar to OCs observed in the fractured scale of *trap:GFP*; *osterix: mCherry* double-transgenic animals (shown in Fig. 2e).

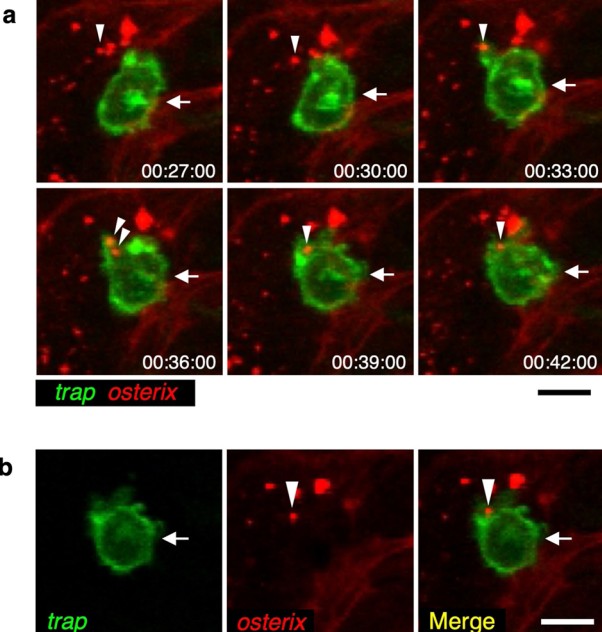

**Fig. 4 trap:GFP+ OCs engulf OB-derived EVs. a** Representative time-lapse imaging of the fractured scale in a *trap:GFP; osterix:mCherry* double-transgenic animal. Six sequences from Supplementary Movie 6 are presented, documenting the stepwise engulfment of OB-derived EVs (arrowheads) by a *trap:GFP+* cell (arrows). **b** Single plane of a z-stack of a *trap:GFP+* cell at the time point of 39 min. Images show a green (*trap:GFP*), red (*osterix:mCherry*), and merged channel. Bars, 10 μm. Experiments were performed six times with similar results.

To clarify the role of OB-derived EVs, KMCs were further cultured with or without OB-derived EVs (mCh+ Hoe^low). Although the number of GFP+ cells increased, the total number of KMCs was unchanged by treatment with OB-derived EVs (Supplementary Fig. 4a, b), suggesting that treatment with EVs does not affect hematopoietic cell proliferation, but does affect OC differentiation from hematopoietic cells. In addition, the percentage of multinucleated GFP+ cells significantly increased by treatment with EVs, but the percentage of mononucleated GFP+ cells decreased. We also observed a few GFP+ cells with more than three nuclei in KMCs treated with OB-derived EVs (Supplementary Fig. 4c). Taken together, these data suggest that treatment with OB-derived EVs promotes differentiation and fusion of OCs in vitro.

Since the high level of *rankl* mRNA was detected in OB-derived EVs, we next questioned if OB-derived EVs mediate Rankl signals to promote OC differentiation. We utilized a gene knockdown method based on the CRISPR/Cas9 system, in which injection of multiple guide RNAs (gRNAs) redundantly targeting a single gene leads to a high proportion of null phenotypes at the G0 generation[40]. Four different gRNAs targeting either exon 1 or 4 of *rankl* gene were co-injected with Cas9 protein into one-cell stage embryos from *trap:GFP* and/or *osterix:mCherry* zebrafish (Fig. 7f). Embryos injected with *rankl* gRNA could partially survive into adulthood. However, most adult animals showed severe body curvature (Supplementary Fig. 5a). Efficiency of *rankl* knockdown was examined in the adult fin by qPCR using two different sets of primers that recognize gRNA target sites of *rankl* gene. We detected ~60–90% reduction of *rankl* expression by each primer set in all seven individual animals tested (Fig. 7f, g), suggesting that wild type *rankl* expression largely decreased in *rankl* gRNA-injected adult animals. The scale formation was,

however, nearly unaffected in *rankl* gRNA-injected animals in terms of the size and morphology (Supplementary Fig. 5b). In mice, deficiency of *Rankl* results in the reduction of multi-nucleated OCs in bone tissue[41]. Similar to *Rankl*-deficient mice, *rankl* gRNA-injected zebrafish showed reduction in both the percentage and absolute number of *trap:GFP*^high OCs compared with wild type zebrafish. In contrast, the number of OBs (GFP^- mCh+ Hoe^high) and OB-derived EVs (mCh+ Hoe^low) was unaffected in *rankl*-gRNA-injected zebrafish (Supplementary Fig. 5c–e), suggesting that knockdown of *rankl* does not affect secretion of EVs in OBs. We co-cultured KMCs from wild type *trap:GFP* animals with OBs (mCh+ Hoe^high) or EVs (mCh+ Hoe^low) from wild type or *rankl* gRNA-injected *osterix:mCherry* animals. We found that the number of GFP+ cells significantly decreased in KMCs co-cultured with EVs from *rankl* gRNA-injected animals compared with those from wild type animals, whereas it was nearly unchanged between KMCs co-cultured with OBs from wild type and *rankl*gRNA-injected animals (Fig. 7h). Taken together, these data suggest that the uptake of OB-derived EVs promotes the differentiation of OCs in a Rankl signaling-dependent manner.

## Discussion

In the present study, we have developed a fracture healing model using double-transgenic zebrafish scales to visualize and isolate OCs and OBs. Our data showed that immature OCs engulf OB-derived EVs prior to convergence at the fracture site. Moreover, co-culture of hematopoietic cells with OB-derived EVs promoted the differentiation of OCs via Rankl signaling. These findings provide insights into the fundamental regulatory mechanisms of OC differentiation by OBs in bone tissue.

Many metabolic or genetic bone diseases are associated with the disruption of the intercellular communication between OCs and OBs[17]. Regulation of OC differentiation by OBs has been shown by many in vitro studies in mice. Monocyte/macrophage precursors can give rise to OCs in the presence of M-CSF and RANKL. M-CSF is secreted in part by OBs and binds to its receptor, c-Fms, expressed on OCs to regulate the survival, migration, and bone resorption activity in OCs[19]. RANKL–RANK signaling activates its downstream targets, including the TNF receptor-associated factor (TRAF) family, which regulates OC formation, survival, and activation via mul-tiple signaling pathways[42]. In vitro studies have thus elucidated roles of various signaling molecules that regulate OC differ-entiation and function. It is still challenging, however, to inves-tigate OC–OB interactions in vivo, and hence the cellular and molecular mechanisms involved in cell–cell communication between these two types of cells remain largely elusive. Ishii's group applied intravital two-photon imaging, which allows to visualize the behavior of living OCs in mouse bone, and detected dynamic communication between OCs and OBs. They described two subsets of functional OCs in terms of their motility and function, 'static—bone resorptive' and 'moving—non-resorptive', which can be shifted by a direct contact with mature OBs[43,44]. In the medaka fin model, it has also been shown that there are two types of OCs in the amputated fin, early-induced and late-induced OCs. The former was relatively small with low TRAP-activity and resorbed bone fragments, while the late-induced OCs were large with high TRAP activity and remodeled the callus[45]. These imaging analyses provided insight into the functional divergence of OCs, highlighting the importance of in vivo live-imaging analysis to better understand OC and OB functions. We could also distinguish at least three different types of OCs and OBs based on morphology in the fractured scale. While further work is needed to precisely describe the maturation process of

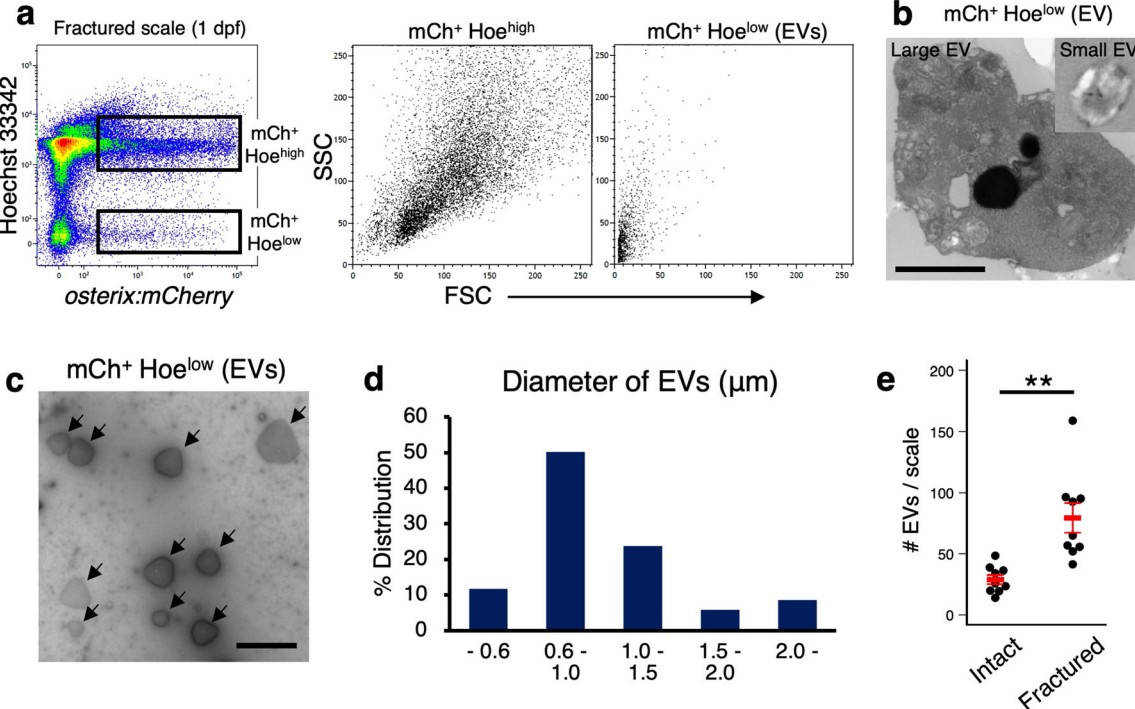

**Fig. 5 Isolation of OB-derived EVs. a** Representative flow cytometric analysis of cells in fractured scales at 1 day post-fracture (dpf) from an *osterix:mCherry* single-transgenic zebrafish. Gated regions in the left panel indicate the *osterix:mCherry*+ Hoecht 33342high (mCh+ Hoehigh) cell fraction and mCh+ Hoelow EV fraction. mCh+ Hoehigh cells and mCh+ Hoelow EVs are displayed in a forward scatter (FSC) vs. side scatter (SSC) dot plot (middle and right panels, respectively). **b** Representative electron microscopic images of isolated large and small EVs. **c** Representative image of isolated EVs negatively stained. Arrows indicate an EV. **d** Percent size distribution of EVs ($n = 223$). **e** Absolute number of mCh+ Hoelow EVs in an intact and fractured scale at 1 dpf. **$p < 0.01$ ($n = 9$ for each group). Bars, 1 μm **b**; 2 μm **c**. Experiments were performed twice with two biological replicates **a–d** and nine biological replicates **e** in each group.

these cell types, these morphological features may reflect different stages of OBs and OCs. Our model system using the transgenic zebrafish scale enables visualization of the whole tissue at a single cell level throughout the process of bone resorption and formation. Indeed, we have successfully captured the moment of cell–cell fusion between two OCs as well as the convergence of OCs at the fracture site, as has been shown in mammalian OCs[46,47]. In addition, the recent advent of gene knockout/knockdown methods in zebrafish based on the CRISPR/Cas9 system[40,48,49] will extend our scale model to rapid and scalable mutagenesis strategies to further probe the function of OCs and OBs. Our imaging strategy in the zebrafish scale will thus open new avenues to elucidate molecular cues needed to regulate OC–OB communication.

The availability of double-transgenic zebrafish, *trap:GFP*; *osterix:mCherry*, also allowed us to investigate the role of EVs in bone tissue. Different subtypes of EVs have been increasingly recognized as potent vehicles of intercellular communication in the past decade. Studies in mammals revealed that EVs released by healthy and apoptotic cells transport biologically active molecules, such as lipids, proteins, mRNAs, and microRNAs to target cells; however, our current knowledge of EVs in the diversity, internalization, and cargo delivery is still very limited[34]. Recently, it was reported that a zebrafish transgenic line that expresses human CD63 fused with pHluorin under the control of tissue-specific promoter/enhancer can be used to visualize the release and uptake of EVs in vivo. A role for EVs in the formation of metastatic niches in vivo could also be determined by tracking EVs shed from melanoma cells in zebrafish embryos, demonstrating that zebrafish is an excellent model for the study of EVs[50]. In the present study, our transplantation assays clearly

demonstrated that mCherry+ particles observed in *trap:GFP*+ cells are derived from EVs shed from OBs. Our FCM analysis revealed that ~75% of *trap:GFP*high cells were detected in the mCherry+ fraction at 1 dpf, suggesting that the majority of mature OCs experience uptake of OB-derived EVs during their differentiation process. Interestingly, our cell culture assays showed that uptake of ABs also promotes OC differentiation in KMCs, supporting the hypothesis that accumulation of apoptotic OBs/osteocytes induces the recruitment of OCs to initiate bone resorption[51]. Since OCs arise from monocyte/macrophage precursors, it is likely that phagocytosis of apoptotic OBs also triggers the differentiation of OCs. In mammals, EVs derived from OCs also play an important role in osteoclastogenesis as well as osteoblastogenesis. EVs from OC precursors promote OC formation in whole bone marrow cultures, whereas EVs from OC-enriched cultures inhibit osteoclastogenesis[26]. Moreover, EVs released by mature OCs contain a high level of RANK and promote bone formation by triggering RANKL reverse signaling in OBs[27]. These observations strongly suggest that cell–cell communication in osteoclastogenesis and osteoblastogenesis is largely dependent on the release and uptake of EVs. Such EV-mediated intercellular communication represents a previously unrecognized cellular mechanism in the bone. Further analysis of EVs in bone tissue will elucidate the molecular mechanisms that regulate the balance between bone resorption and formation.

## Methods

**Zebrafish husbandry and fracture stimulation**. Zebrafish were raised in a circulating aquarium system (AQUA) at 28.5 °C in a 14/10 h light/dark cycle and maintained in accordance with guidelines of the Committee on Animal Experimentation of Kanazawa University. For fracture stimulation, adult zebrafish were anesthetized in system water containing 0.01% tricaine (Sigma), and the epidermis

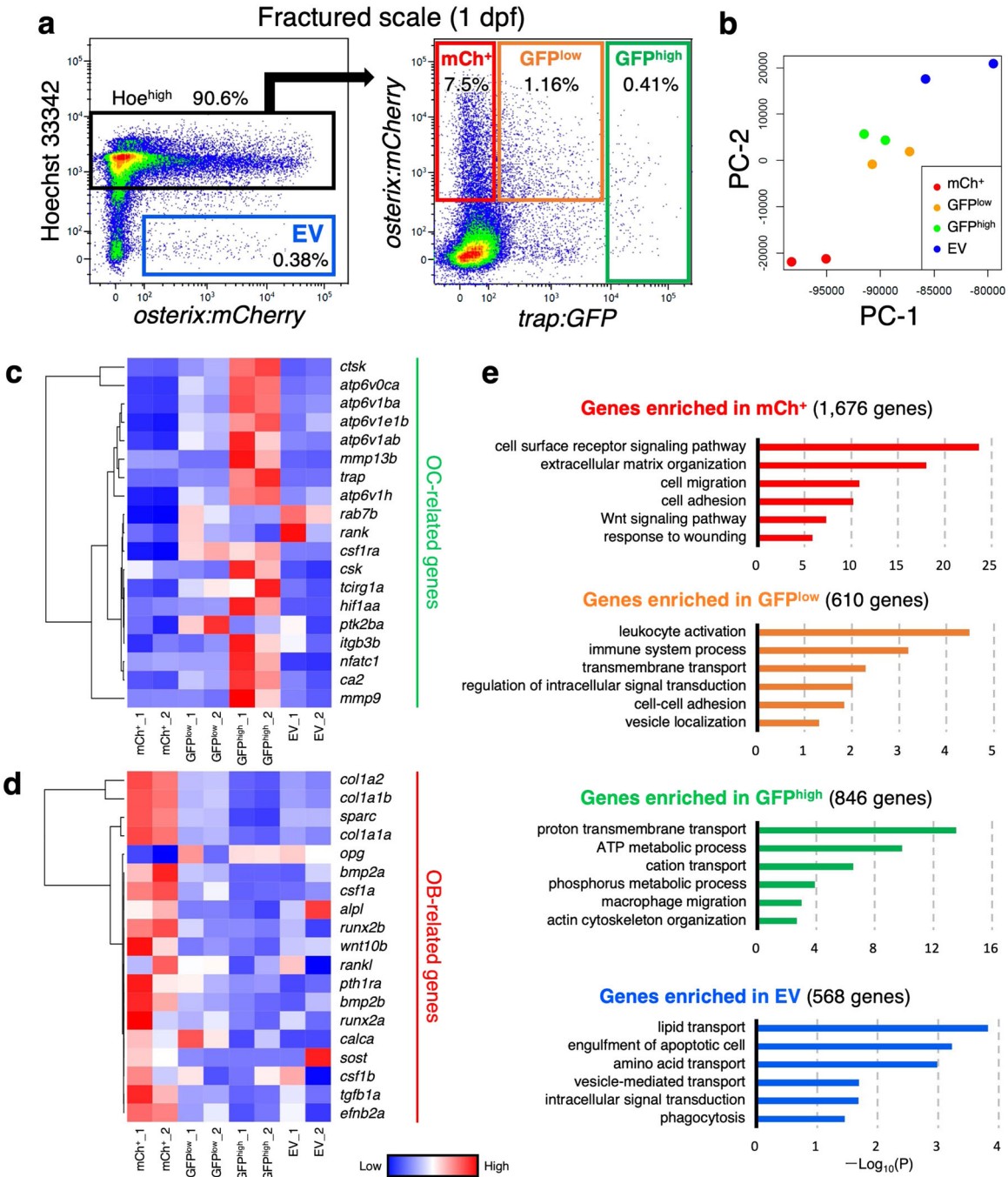

**Fig. 6 Transcriptome analysis of OBs, OCs, and OB-derived EVs. a** Representative flow cytometric analysis of cells in scales at 1 day post-fracture (dpf) from a *trap:GFP; osterix:mCherry* double-transgenic animal. Gated regions in the left panel indicate the Hoechst 33342[high] (Hoe[high]) cell fraction and *osterix: mCherry*[+] Hoe[low] fraction ("EV"). Cells in the Hoe[high] fraction are displayed in the right panel to further divide into three populations, *trap:GFP*[−] *osterix: mCherry*[+] ("mCh[+]"), *trap:GFP*[low] *osterix:mCherry*[+] ("GFP[low]"), and *trap:GFP*[high] ("GFP[high]"). **b** Principal component analysis (PCA) based on the read per million (RPM) of each sample. **c, d** Hierarchical clustering of selected OC-related **c** and OB-related genes **d** in the mCh[+], GFP[low], GFP[high], and EV fraction in the fractured scale. **e** Gene ontology enrichment analysis of highly expressed genes in the mCh[+], GFP[low], GFP[high], and EV fraction. Experiments were performed once with two biological replicates.

area of a scale was cut ~400 μm in length with fine scissors under a fluorescent stereo microscope (Axiozoom V16, Zeiss).

**Generation of transgenic lines**. Zebrafish transgenic lines of *Tg(trap:GFP-CAAX)*[ou2031Tg] and *Tg(osterix:Lifeact-mCherry)*[ou2032Tg] were generated as previously described[1,2]. Briefly, a 6-kb upstream regulatory region of the zebrafish *trap* (*acp5a*) gene and 4.1 kb upstream regulatory region of the medaka *osterix* (*sp7*) gene were amplified using the primers listed in Supplementary Table 1 and ligated into pT2AL200R150G containing *GFP-CAAX* and *Lifeact-mCherry*, respectively.

*TPase* mRNA was synthesized with the *pCS2+* vector by mMESSAGE mMA-CHINE SP6 Transcription Kit (Thermo Fisher Scientific). Transgenic zebrafish were generated by injection of the plasmid construct with *TPase* mRNA into 1-cell stage embryos. A stable transgenic line was obtained by screening of GFP or mCherry expression in F1 generation.

**Knockdown of *rankl* by CRISPR/Cas9 system**. Previously designed guide RNA (gRNA) sequences targeting the *rankl* exon 1 or 4 were utilized[40]. gRNAs were synthesized as previously described[40]. Briefly, a single-strand DNA oligo

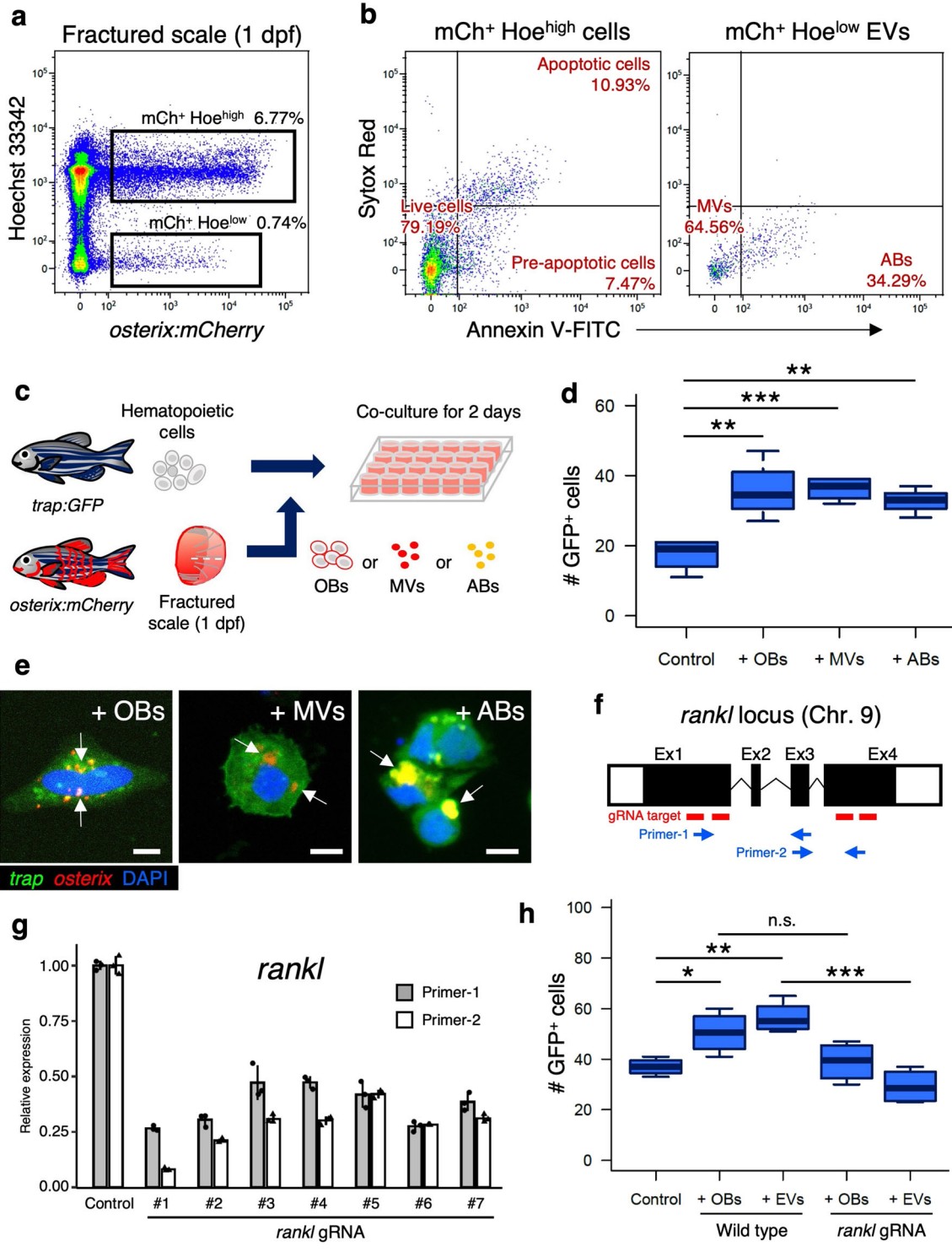

containing the gRNA target and T7 promoter sequence was annealed with a gRNA scaffold primer. Double-strand DNAs were then synthesized with MightyAmp DNA polymerase (Thermo Fisher Scientific), followed by in vitro transcription using MEGAshortscript T7 Transcription Kit (Thermo Fisher Scientific). gRNAs were purified with mirVana miRNA Isolation Kit (Thermo Fisher Scientific) according to the manufacture's protocol. Embryos from *trap:GFP* and/or *osterix: mCherry* animals were injected with a mixture of Cas9 protein (400 ng/μL) (IDT) and gRNAs (100 ng/μL each) at one-cell stage. gRNA sequences and gRNA scaffold primer are listed in Supplementary Table 1.

**Cell preparation and flow cytometry.** Scales were extracted from anesthetized zebrafish under a stereo microscope, and were treated with Liberase TM (Roche) in

PBS for 1 h at 37 °C. Cells were then filtered through 40 μm stainless mesh and washed with ice-cold 2% fetal bovine serum (FBS) in Hanks' balanced salt solution (HBSS, Wako) by centrifugation (300×g). KMCs were prepared as previously described[52] with some modifications. Briefly, cells were obtained by pipetting of a dissected kidney in 1 mL of ice-cold 2% FBS in HBSS. After centrifugation, the pellet was gently mixed with 1 mL of distilled water by pipetting to lyse erythrocytes by osmotic shock. Subsequently, 1 mL of 2X HBSS was added. Cells were then filtered through a 40 μm stainless mesh and washed with 2% FBS in HBSS by centrifugation. For staining with Hoechst 33342 (Hoe), cells were resuspended at a density of $10^6$ cells/mL in 2% FBS in HBSS and were stained with 5 μg/mL Hoe (Thermo Fisher Scientific) for 90 min at 25 °C in dark with gentle agitation. For staining with Annexin-V, cells were washed with Cell Staining Buffer (BioLegend)

**Fig. 7 OB-derived EVs promote OC differentiation via Rankl signaling. a** Representative flow cytometric analysis of cells in scales at 1 day post-fracture (dpf) from a *osterix:mCherry* single-transgenic animal. Gated regions indicate the mCh⁺ Hoe^high^ cell fraction and mCh⁺ Hoe^low^ EV fraction. **b** mCh⁺ Hoe^high^ cells and mCh⁺ Hoe^low^ EVs were displayed in an Annexin-V-FITC vs. Sytox Red dot plot. mCh⁺ Hoe^high^ cells were subdivided into three fractions, "live", "pre-apoptotic", and "apoptotic", whereas mCh⁺ Hoe^low^ EVs were divided into two fractions, "microvesicle" (MV) and "apoptotic body" (AB). **c** Schematic diagram of in vitro cell culture assays. Kidney marrow cells (KMCs) from *trap:GFP* single-transgenic zebrafish were co-cultured with OBs, MVs, or ABs from fractured scales of *osterix:mCherry* single-transgenic zebrafish in a fibronectin-coated plate. At 2 days of co-culture, the number of *trap:GFP*⁺ cells was counted in each well. Non-co-cultured KMCs were used as a control. **d** The average number of *trap:GFP*⁺ cells in each type of wells. Error bars, s.d. ($n = 4$ for each group). **e** Representative images of *trap:GF*P⁺ cells co-cultured with OBs (left panel), MVs (middle panel), or ABs (right panel). *trap:GFP*⁺ cells contained OB-derived EVs in the cytoplasm (arrows). Bars, 5 μm. **f** Schematic diagram of zebrafish *rankl* locus. gRNA target sites and primer recognition sites are shown in red bars and blue arrows, respectively. **g** qPCR analysis of *rankl* in the adult fin of wild type control and *rankl* gRNA-injected animals. Results from seven individual gRNA-injected animals are shown. Data are mean ± s.d. from three independent experiments. **h** The average number of *trap: GFP*⁺ cells in non-co-cultured KMCs (control), or KMCs co-cultured with OBs or EVs from wild type or *rankl* gRNA-injected animals. Error bars, s.d. ($n = 4$ for each group); n.s., no significance; *$p < 0.05$; **$p < 0.01$; ***$p < 0.001$ by one-way ANOVA followed by Dunnett's test. Experiments were performed twice with four biological replicates in each group **d**, **h**.

by centrifugation, resuspended in Annexin-V-binding buffer containing Annexin-V-FITC (BioLegend), and stained for 15 min at room temperature in dark. Just before flow cytometric (FCM) analysis, the Sytox Red (Thermo Fisher Scientific) was added at a concentration of 5 nM to exclude dead cells or to detect apoptotic cells. FCM acquisition and cell sorting were performed on a FACS Aria III (BD Biosciences). Data analysis was performed using the Kaluza software (ver. 1.3, Beckman Coulter). Since the detectable size of cells/particles by FACS Aria III is more than 0.5 μm, it is not possible to separate very small EVs (<0.4 μm), such as exosomes, in this analysis. EVs were therefore isolated according to the same procedure as cells to minimize contamination with EVs of unknown origin. The absolute number of cells was calculated by flow cytometry based on acquisition events, maximum acquisition times, and the percentage of each cell fraction.

**Cell culture**. KMCs from *trap:GFP* animals were resuspended in medium containing 40% Leiboviz's L-15 medium (Wako), 32% Dulbecco's modified Eagle's medium (Wako), 12% Ham's F12 medium (Wako), 8% FBS, 2 mM ʟ-glutamine (Wako), 15 mM 4-(2-hydroxyethyl)-1-piperazineethanesulfonic acid (HEPES, Sigma), 100 U penicillin (Wako), and 100 μg/mL streptomycin (Wako). Approximately $6 \times 10^4$ of KMCs were plated on a 96-well plate coated with fibronectin (Corning) and incubated for 3 h at 30 °C, 5% $CO_2$. Subsequently, ~2000 of sorted cells or EVs were plated in the wells (four wells for each fraction) and incubated for additional 2 days at the same condition described above. The number of GFP⁺ cells was counted using an EVOS FL Cell Imaging System (Thermo Fisher Scientific). For counting the number of nuclei, Hoe was directly added into the medium at a concentration of 5 μg/mL. For counting the total number of cells in each well, cells were treated with 0.25% trypsin–1 mM EDTA and harvested by pipetting. The total number of cells was counted using a hemocytometer (Funakoshi).

**RNA-seq and qPCR**. For sorted cells and EVs, whole-transcript amplification and double-strand cDNA synthesis was performed according to the Quartz-Seq method as previously described[53,54]. Cells were directly sorted in a lysis buffer containing 1 μg/mL of polyinosinic–polycytidylic acid, and total RNA was extracted using RNeasy Mini Kit. Reverse transcription (RT) was performed using Super Script III (Thermo Fisher Scientific) and an RT primer, which contains oligo-dT, T7 promoter, and PCR target region sequences. After digestion of remaining RT primers by exonuclease I (Takara), a poly-A tail was added to the 3′ ends of the first-strand cDNAs using terminal transferase (Sigma). The second-strand DNA was then synthesized using MightyAmp DNA polymerase (Thermo Fisher Scientific) and a tagging primer. PCR amplification was performed using a suppression primer. The amplified double-strand cDNA was purified using QIAquick PCR Purification Kit (Qiagen). Library preparation for RNA-seq was performed using Nextera XT DNA Library Preparation Kit (illumina). Next generation sequencing of cDNA libraries was performed by GENEWIZ using the Illumina NextSeq500 (illumina), and base-calling was performed using the Illumina RTA software (ver. 2.4.11). Sequence reads were mapped to the zebrafish reference genome (GRCz11) using HiSAT2 (version 2.1.0). Reads per million (RPM) were calculated using Subread (ver. 1.6.4). Up-regulated genes of each fraction were selected with a p-value cutoff of 0.05 based on one-way ANOVA and more than two-fold change. PCA and hierarchical clustering of each subset was performed in R (ver. 3.5.0) with the Bioconductor gplots package. The data have been deposited in Gene Expression Omnibus (GEO) (National Center for Biotechnology Information) and are accessible through the GEO database (series accession number, GSE134330). Total RNAs from the adult fin were extracted with RNeasy Mini Kit (QIAGEN) and cDNAs were synthesized with ReverTra Ace qPCR RT Master Mix (Toyobo). Quantitative real-time PCR (qPCR) assays were performed using TB Green Premix Ex Taq II (TaKaRa) on a ViiA 7 Real-Time PCR System according to manufacturer's instructions (Thermo Fisher Scientific). Primers used for whole-transcript amplification and qPCR are listed in Supplementary Table 1.

**Electron microscopy**. The ultrastructure of cells and EVs was observed by transmission electron microscopy as previously described[8] with some modifications. Briefly, sorted cells and EVs were fixed with 2.5% glutaraldehyde (Nacalai Tesque), 2% paraformaldehyde (Wako) in 0.1 M phosphate buffer (pH 7.4) at 4 °C overnight. Cells were then dehydrated and embedded in Epon 812 (TAAB Laboratories). Ultrathin sections were obtained from the Epon blocks and stained with uranyl acetate and lead citrate. For negative staining of EVs, a grid with a supporting film was placed on a drop of the EV suspension for 15 min and excess solution was removed with a filter paper. The grid was then placed in 2% uranium acetate aqueous solution for 2 min, followed by air-drying. Sections and stained EVs were observed under an electron microscope (H-7650, Hitachi).

**X-ray irradiation and transplantation**. Three to six zebrafish were placed in a 90 mm Petri dish in system water, and animals were sublethally irradiated with X-ray on a Faxitron RX-650 (Faxitron, 130 kVp, 1.15 Gy/min) for 20 min (~23 Gy). At 2 days post-irradiation, animals were transplanted with KMCs using a retro-orbital injection method[31].

**Intubation anesthesia and imaging**. Intubation anesthesia was performed as previously described[55] with some modifications. A flask containing 0.04% of 2-phenoxyethanol (Wako) in system water was kept in a water bath to maintain a constant temperature of 28 °C, and delivered to a glass-bottom chamber (Eppendorf) using a peristatic pump (ATTO). An adult zebrafish mounted in the chamber was orally perfused with the anesthetic water using an Intramedic polyethylene tube (BD). Excess water in the chamber was removed and returned to the flask using another peristatic pump. For imaging of extracted scales, scales were stained with 5 μg/mL of Hoe for 20 min at room temperature and mounted in a glass bottom dish (Matsunami) filled with 0.6% low-gelling agarose (Sigma). For imaging of sorted cells, cells were plated on a fibronectin-coated glass-bottom dish with culture medium described above. For imaging of cultured cells, cells were plated on a cover slip coated with fibronectin (Cosmo Bio) and were cultured as described above. Cell were then fixed with 4% paraformaldehyde and mounted with ProLong Gold Antifade Mountant with DAPI (Thermo Fisher Scientific). Fluorescent images were captured using an FV10i confocal microscope and Fluoview FV10i-SW software (ver. 2.1.1) (Olympus). For time-lapse imaging, images were captured every 3 or 5 min for scales and 30 s for cells, and movies were generated using iMovie software (ver. 10.1.12) or Fluoview FV10i-SW software. We defined GFP "bright" as having four-fold higher than the average intensity of GFP⁺ cells in the intact scale and mCherry "bright" as having 10-fold higher intensity than the average mCherry intensity of the whole intact scale. Visible light imaging of adult animals were captured using an Olympus Pen E-PL8 distal camera (Olympus).

**Statistics and reproducibility**. Statistical differences between groups were determined by unpaired two-tailed Student's *t*-test or one-way ANOVA with Dunnett's test. A value of $p < 0.05$ was considered to be statistically significant. All experiments were repeated at least twice with similar results.

**Reporting summary**. Further information on research design is available in the Nature Research Reporting Summary linked to this article.

## Data availability

The raw data of RNA-seq can be obtained from the GEO database (series accession number, GSE134330). The source data for Figs. 2, 5, 7, Supplementary Figs. 2–5 are provided as Supplementary Data 1. All data and materials produced by this study are available from the corresponding author upon request.

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

## Acknowledgements

The authors thank Dr. S. Yuge for supporting intubation anesthesia, Dr. M. Hazawa for supporting cell culture assays, Dr. Y. Furusawa for supporting RNA-seq analysis, and Dr. K. Lewis for critical evaluation of the manuscript. This work was supported in part by Grant-in-Aid for Scientific Research (C) from the Japan Society for the Promotion of Science (18K06331).

## Author contributions

J.K.-S., M.K., S.Y., J.K., and M.I. performed experiments; J.K.-S., N.S., K.K., A.H., and I.K. discussed results; N.S., K.K., A.H., and M.Y. edited the manuscript; J.K.-S. and I.K. designed the research, analyzed data, and wrote the manuscript.

## Competing interests

The authors declare no competing interests.
