## [Peer Review File · Communications Biology]

Reviewers' comments:

Reviewer #1 (Remarks to the Author):

The paper describes studies in the zebrafish scale, showing aspects of communication between osteoblasts and osteoclasts. Genetic methods have been used to label each of the two cell classes, and a fracture model used to study their behaviour, using imaging techniques. This is interesting paper work providing evidence in the fish scale of OB- derived extracellular vesicles being engulfed by OCs and perhaps contributing to their differentiation. The latter is not new for mammalian cells, but the major interest of this paper comes from the comparative biology point of view, in showing in such detail cell communication processes that might be very similar to those in mammalian cells. It illustrates the conservation of process among vertebrates, and given that so little is generally appreciated about the biology of the fish scale, this is quite valuable.

Specific matters

1. I suggest that the introduction needs extensive rewriting, with the authors concentrating on the cell communication processes in the fish scale that they illustrate so well in the paper. The authors at several points use the terms "modelling" and "remodelling" to describe the cellular events. but there is nothing in the present paper to relate in any way to either of these processes, or indeed that the terms are relevant for the fish scale. There is very real interest in the findings, but that interest relates to intercellular communication processes between osteoclastic and osteoblastic lineage cells within the fish scale.

2. Introduction page 3, paragraph 2, refers to "recent studies showing EVs can communicate between osteoblasts and osteoclasts". The concept of EV signalling among bone cells is a recent one, and there are relevant and important references that should be used for this statement. The authors use reference 9 but this reviewer found ref 9 impossible to locate. It is probably not an appropriate review to be used as a reference, and certainly not helpful if it is that difficult for the reader to find. There are much more important references – especially Hunyh et al (J Dent Res 95:673, 2016) and Cappiarello et al (ref 15, JBMR 2018) who showed that RANKL- containing SEVs from OBs increase OC formation. Especially important is the paper by Ikebuchi et al (Nature 2019). The present work in the fish scale complements the latter work very well and it is important to discuss this from the comparative biology point of view. Otherwise this paper might be looked at as containing nothing novel. My point as a reviewer is that its novelty lies in its demonstration of SEVs and their function in cells of the fish scale.

3. With regard to novelty, the authors emphasise the fact that they have demonstrated fusion of cells within the osteoclast lineage. It is true that is the first time it is shown in fish cells as far as this reviewer is aware, but there is a very extensive literature on the process of osteoclast fusion in mammalian cells (e.g. Xing et al, World J Orth. 3:212, 2012; Soe et al, Bone 73: 181, 2015; Levaot et al, Bone 79: 21, 2015; Jacome-Galarza et al, Nature 568: 541, 2019...etc)

4. Page 3, paragraph 2, lines 3 – 4. RANKL-RANK Communication is of course accepted as important. The Eph signalling component (refs 5-7) should be omitted. This is quite disputed as a coupling factor as claimed in the reference 5.

5. Results, figure 1 and elsewhere. How were the decisions made to define cells as cherry or GFP "bright" and "low"? This seems subjective

6. Results page 4, paragraph 2, line 1. " to examine the process of bone remodelling...". This should be omitted. Remodelling refers to a specific set of events in which resorption of a certain amount of

bone is followed by a reversal phase then a formation phase during which the resorbed space is filled by differentiating osteoblasts (there are very many reviews of this process and the authors must be aware of this). As noted above, there is nothing in this paper, or elsewhere as far as this reviewer knows, to warrant that use of "remodelling" to describe what is happening in the fish scale. The authors repeat this on page 9 (middle), where they even write of "bone remodelling at the single cell level". It is difficult to know what they could mean by that.

7. Results page 5 and figure 2. The "three different types of OB" (based on shape) seem most likely to reflect different stages of OB differentiation, and the same can be said of the different shape OCs. This should be discussed by the authors. It is one of the limitations of their methods, in that they cannot identify stages of osteoblast or osteoclast differentiation with any confidence. These criticisms do not invalidate the work but they should be acknowledged.

8. The transplantation experiments provide useful information, as also does the fracture model showing communication among the cells. The conclusions from these experiments might best be modified though. While the uptake of cherry+ve components into OC lineage cells is shown convincingly, evidence for promotion of OC differentiation is lacking. The important fact remains that strong evidence is presented for the release of EVs by OB cells of the scale and the uptake by OC lineage cells. This might contribute to differentiation but this has not been directly shown.

Reviewer #2 (Remarks to the Author):

In this article, Kobayashi-Sun and colleagues describe the interaction between osteoclast and osteoblast in vivo. Using double transgenic zebrafish, labelling both cell types, they develop a model of scale injury leading to osteoclast recruitment. Strikingly, they observe that most OCs contain OB labelling. They further provide some evidences, including nice intravital imaging, that this exchange could be mediated by extracellular vesicles (EVs) and that it could promote OCs differentiation through Rankl signaling.

This article describes a powerful model to image and study OBs/OCs interactions in vivo. However, it requires additional experiments to prove its main findings.

Major points:

- The population of EVs sorted by FACS needs to be better characterized. It should be analysed by electron microscopy and nanoparticle tracking analysis (or similar approach) to prove their vesicular nature and their size distribution. In addition, EVs should be isolated directly from sorted and in vitro cultured populations of OBs and OCs by standard methods (Ultracentrifugation, density gradients) and characterized by EM and NTA. This is important, as it will allow to identify populations of EVs which are detected by flow cytometry (related to that, the minimal size of objects detected by flow cytometry should be notified) but still are likely to exist in vivo and contribute to OBs/OCs communication.
- Similarly, the functional experiments presented in figure 6 are essential and should be complemented to fully prove the role of EVs. First, the phenotypes should be more detailed: why are there more GFP+ cells after treatment? Increased proliferation or decreased cell death ? Other phenotypes, such as cell migration, cell fusion (by measuring the proportion of multinucleated cells for instance) and differentiation should be quantified. Finally, EVs isolated from in vitro cultured OBs as mentioned in the first point should be tested on OCs.
- Another important statement of the paper is the involvement of Rank signalling in the differentiation of OCs. Here the authors show the expression of Rank and Rankl transcripts in EVs isolated by flow cytometry and use a CRISPR-induced KO of Rank in the whole fish to test its role in OCs

differentiation. Alternative approaches should be used to 1) really prove the presence of Rankl protein on OBs EVs surface and 2) properly demonstrate the function of EV-Rankl in OCs differentiation. Could the authors test if any available anti-mammalian Rankl antibody works on zebrafish ? In addition, it should be determined whether rankl KO affects EV secretion.

Other points:

- Some phenotypes should be better described. For instance, the proportion of the different types of OBs and OCs observed in culture (figure 2) should be quantified, before and after scale fracture. The in vivo movies would also benefit from some quantifications: for instance, what is the concentration of EVs with and without scale fracture ?
- Some experimental details are lacking. For instance, in the very nice movie, dissected in figure 4, what type of imaging are we looking at ? If this is confocal, is it single planes or Z-projections ? If this is not confocal, how can the authors make sure that the particles have been internalized ? Another example is the functional experiments presented in figure 6: how are the experiment normalized ? are similar numbers of cells/EVs added to OCs, or similar amounts of proteins ? This should be specified.
- Description of the compartments storing OBs particles within OCs cells is an important aspect. Authors could co-stain for late endosomes/lysosomes, for instance using lysotracker probes. It could allow to identify differences in the uptake of MVs or Abs (figure 6).
- In figure 2F, authors show an EM picture with arrows pointing to "lysosomes" ; these compartments are clearly a mixture of different types, including early endosomes and multi-vesicular bodies. More convincing images should be presented or the legends should be modified.
- In the discussion, the section dedicated to the study of EVs in vivo should be more detailed, as this paper presents valuable novel model and data to the field. In particular, recent papers describing the use zebrafish in EVs tracking should be cited (reviewed in Verweij et al. Trends in Cell Biology 2019).

Responses to Reviewers (Kobayashi-Sun et al., *Communications Biology*)

We thank the reviewers for their many helpful suggestions. We believe the manuscript is now greatly improved by the inclusion of new data generated in response to reviewer comments.

Reviewer #1

1. I suggest that the introduction needs extensive rewriting, with the authors concentrating on the cell communication processes in the fish scale that they illustrate so well in the paper. The authors at several points use the terms “modelling” and “remodelling” to describe the cellular events. but there is nothing in the present paper to relate in any way to either of these processes, or indeed that the terms are relevant for the fish scale. There is very real interest in the findings, but that interest relates to intercellular communication processes between osteoclastic and osteoblastic lineage cells within the fish scale.

Thank you very much for pointing out problems in the introduction. As the reviewer mentioned, the term “remodeling” was misused in our previous version, and we have omitted this word from the revised manuscript. According to the reviewer’s suggestions including comments below, we have greatly revised the introduction. It should be noted, however, that our present study focuses on cellular communication in OC differentiation, particularly the role of OB-derived EVs. The revised version of the introduction is therefore focused mainly on our current knowledge of cellular communication involved in OC differentiation.

2. Introduction page 3, paragraph 2, refers to “recent studies showing EVs can communicate between osteoblasts and osteoclasts”. The concept of EV signalling among bone cells is a recent one, and there are relevant and important references that should be used for this statement. The authors use reference 9 but this reviewer found ref 9 impossible to locate. It is probably not an appropriate review to be used as a reference, and certainly not helpful if it is that difficult for the reader to find. There are much more important references – especially Hunyh et al (J Dent Res 95:673, 2016) and Cappiarello et al (ref 15, JBMR 2018) who showed that RANKL- containing SEVs from OBs increase OC formation. Especially important is the paper by Ikebuchi et al (Nature 2019). The present work in the fish scale complements the latter work very well and it is important to discuss this from the comparative biology point of view. Otherwise this paper might be looked at as containing nothing novel. My point as a reviewer is that its novelty lies in its demonstration of SEVs and their function in cells of the fish scale.

Thank you again for noting improper citations. As the reviewer suggested, we have omitted reference #9 and added new references that are more relevant to the background of our study.

3. *With regard to novelty, the authors emphasise the fact that they have demonstrated fusion of cells within the osteoclast lineage. It is true that is the first time it is shown in fish cells as far as this reviewer is aware, but there is a very extensive literature on the process of osteoclast fusion in mammalian cells (e.g. Xing et al, World J Orth. 3:212, 2012; Soe et al, Bone 73: 181, 2015; Levaot et al, Bone 79: 21, 2015; Jacome-Galarza et al, Nature 568: 541, 2019... etc)*

We have improved the discussion of OC fusion as followed:

Original sentences

Indeed, we have successfully captured the moment of cell-cell fusion between two OCs as well as the convergence of OCs at the fracture site, demonstrating in real time that multinucleated OCs are formed by cell-cell fusion in the fracture site.

Revised sentences

Indeed, we have successfully captured the moment of cell-cell fusion between two OCs as well as the convergence of OCs at the fracture site, as has been shown in mammalian OCs.

4. *Page 3, paragraph 2, lines 3 – 4. RANKL-RANK Communication is of course accepted as important. The Eph signalling component (refs 5-7) should be omitted. This is quite disputed as a coupling factor as claimed in the reference 5.*

We have omitted reference to “Ephrin – Eph signaling” from the introduction.

5. *Results, figure 1 and elsewhere. How were the decisions made to define cells as cherry or GFP “bright” and “low”? This seems subjective*

The intensity of GFP and mCherry is one of the important factors to distinguish cell types in our study. The images shown in Fig. 1c, for instance, have been captured by confocal microscopy at the same condition, including the same laser power and sensitivity. It is possible, therefore, to compare the expression level of transgenes based on the fluorescent intensity among these images. In the revised version of the manuscript, we have defined GFP “bright” as having 4-fold higher than the average intensity of GFP⁺ cells in the intact scale and mCherry “bright” as having 10-fold higher intensity than the average mCherry intensity of the whole intact scale. Based on this definition, only *trap:GFP*⁺ cells observed at the fracture site and *osterix:mCherry*⁺ cells observed at the edge region of the scale were classified as *trap:GFP*^{bright} and *osterix:mCherry*^{bright}, respectively. Other fluorophore-expressing cells were shown as *trap:GFP*⁺ and *osterix:mCherry*⁺ cells (not “low” cells). We have added this definition

to materials and methods.

6. Results page 4, paragraph 2, line 1. “ to examine the process of bone remodelling...”. This should be omitted. Remodelling refers to a specific set of events in which resorption of a certain amount of bone is followed by a reversal phase then a formation phase during which the resorbed space is filled by differentiating osteoblasts (there are very many reviews of this process and the authors must be aware of this). As noted above, there is nothing in this paper, or elsewhere as far as this reviewer knows, to warrant that use of “remodelling” to describe what is happening in the fish scale. The authors repeat this on page 9 (middle), where they even write of “bone remodelling at the single cell level”. It is difficult to know what they could mean by that.

We have omitted the word of “remodeling” in the revised version of the manuscript.

7. Results page 5 and figure 2. The “three different types of OB” (based on shape) seem most likely to reflect different stages of OB differentiation, and the same can be said of the different shape OCs. This should be discussed by the authors. It is one of the limitations of their methods, in that they cannot identify stages of osteoblast or osteoclast differentiation with any confidence. These criticisms do not invalidate the work but they should be acknowledged.

We appreciate this suggestion and agree that it is difficult to determine the differentiation process of OBs and OCs by morphology. According to the reviewer’s suggestion, we have added sentences to discussion as followed:

“We could also distinguish at least three different types of OCs and OBs based on morphology in the fractured scale. While further work is needed to precisely describe the maturation process of these cell types, these morphological features may reflect different stages of OBs and OCs.”

8. The transplantation experiments provide useful information, as also does the fracture model showing communication among the cells. The conclusions from these experiments might best be modified though. While the uptake of cherry+ve components into OC lineage cells is shown convincingly, evidence for promotion of OC differentiation is lacking. The important fact remains that strong evidence is presented for the release or SEVs by OB cells of the scale and the uptake by OC lineage cells. This might contribute to differentiation but this has not been directly shown.

The purpose of the transplantation assays was to demonstrate uptake of OB-derived EVs by *trap:GFP*⁺ OCs in vivo, not to show the role of EVs in OC differentiation. Indeed, we did not mention the function

of EVs in discussing the results of the transplantation assays (Page 5-6, “Transplantation assays confirm uptake of OB-derived EVs in OCs”). Since it is difficult to show the function of EVs by transplantation assays, we utilized in vitro cell culture assays instead to demonstrate the role of EVs in OC differentiation (Fig. 7 and Supplementary Fig. 4).

Reviewer #2

- The population of EVs sorted by FACS needs to be better characterized. It should be analysed by electron microscopy and nanoparticle tracking analysis (or similar approach) to prove their vesicular nature and their size distribution. In addition, EVs should be isolated directly from sorted and in vitro cultured populations of OBs and OCs by standard methods (Ultracentrifugation, density gradients) and characterized by EM and NTA. This is important, as it will allow to identify populations of EVs which are detected by flow cytometry (related to that, the minimal size of objects detected by flow cytometry should be notified) but still are likely to exist in vivo and contribute to OBs/OCs communication.

We appreciate these interesting suggestions and agree on the importance of further characterization of OB-derived EVs. EVs are classified into three types according to their size and origins: exosomes, microvesicles (MVs), and apoptotic bodies (ABs). Due to their size, MVs and ABs, but not exosomes (< 100 nm), can be detected and sorted by our flow cytometer (BD FACS Aria III), which can detect cells/particles more than 500 nm in diameter. Similar to flow cytometric analysis, MVs and ABs can be resolved by our confocal microscope (Olympus FV10i), enabling in vivo live-imaging analysis of EVs together with OBs and OCs. Taking advantage of isolation and visualization, therefore, our present study focused on MVs and ABs to demonstrate that OB-derived EVs play an important role in OC differentiation.

To further characterize EVs, OB-derived EVs (mCh⁺ Hoe^{low}) were sorted from fractured scales and analyzed by electron microscopy (EM). A representative ultrastructural image of OB-derived EVs has been added. In addition, negative staining followed by EM analysis has determined the size distribution of EVs: the majority of EVs (74%) were 0.6 – 1.5 μm in diameter. We believe that our data are sufficient to prove the presence of OB-derived EVs in the zebrafish scale. It is also interesting idea to isolate EVs from OCs in the zebrafish scale. However, our current study focused on the role of OB-derived EVs in OC differentiation, and unfortunately, EVs from OCs are far beyond the scope of this study. We have added these data to a new Fig. 5.

As the reviewer mentioned, isolation of EVs, particularly exosomes, has usually been achieved by ultracentrifugation or density gradients. While these methods can also be applied to isolate EVs directly from zebrafish scales, fractions isolated by these methods would contain very small EVs, including exosomes, the origin of which cannot be determined by flow cytometry due to the size

limitation described above. To minimize contamination with EVs of unknown origin, we isolated EVs from scales following the same procedure used to harvest cells from the scale, including the same centrifugation speed and time (1,600 rpm, 7 min). We then used *osterix:mCherry* and Hoechst staining to distinguish OB-derived EVs (mCh⁺ Hoe^{low}) from OBs (mCh⁺ Hoe^{high}) or EVs from other cell types (mCh⁻ Hoe^{low}), validating that FACS-separable EVs can be isolated without ultracentrifugation or density gradients (Response Fig. 1). After collection of the cell fraction (“Cells”) by centrifugation at 1,600 rpm (300X g), the first supernatant (“Sup1”) was further centrifuged at 4,000 rpm (1700X g) to collect the second supernatant (“Sup2”), which is frequently used for ultracentrifugation or density gradients to isolate exosomes (Crescitelli et al., *J Extracell Vesicles*. 2013; Kowal et al., *PNAS*. 2016). Our data showed that the number of mCherry⁺ EVs was much lower in “Sup2” compared with “Cells”, indicating that FACS-separable EVs can be collected by normal centrifugation speeds. We have added the following sentences to materials and methods:

“Since the detectable size of cells/particles by FACS Aria III is more than 0.5 μm , it is not possible to separate very small EVs (< 0.4 μm), such as exosomes, in this analysis. EVs were therefore isolated according to the same procedure as cells to minimize contamination with EVs of unknown origin.”

Response Fig. 1. Collection of EVs from zebrafish scales.

The absolute number of OB-derived EVs in a scale was much higher in “Cells” compared with that in “Sup2”. *** $p < 0.001$.

Since cultured cells frequently used to obtain a large number of pure EVs for drug delivery or therapeutic investigations (Andaloussi et al., *Nat Rev Drug Discov*. 2013), we attempted to isolate EVs using a similar approach. We examined EVs from cultured OBs to determine if they would also promote OC differentiation, as the reviewer suggested. Unexpectedly, however, it was difficult to isolate EVs from cultured OBs in zebrafish. As shown in Response Fig. 2, mCh⁺ Hoe^{high} OBs (63,000 cells) were directly sorted from fractured scales and cultured for 7 days in a standard OB culture medium (10% FBS in DMEM) (Cappariello et al., *J Bone Miner Res*. 2018; Bertin et al., *Histochem Cell Biol*. 2015; Davies et al., *Front Bioeng Biotechnol*. 2019). Cells were then harvested and re-stained with Hoechst 33342 to isolate EVs by flow cytometry. Although the large fraction of mCh⁺ Hoe^{low} EVs was detected, treatment of hematopoietic cells with these EVs did not affect the number of

trap:GFP⁺ OCs. One of the problems in this experiment is the low survival rate of OBs in vitro. The number of live OBs at 7 days post-culture was 3,500 cells, only 5.5% of the initial OB input, suggesting that the mCh⁺ Hoe^{low} fraction mainly contains cell debris rather than EVs. Therefore, it is currently difficult for us to evaluate the function of EVs from cultured OBs.

Response Fig. 2. Isolation of EVs from cultured OBs.

OBs isolated from fractured scales were cultured for 7 days. EVs (mCh⁺ Hoe^{low}) were then sorted and used for co-culture experiment with hematopoietic cells. There was no significant difference in the number of GFP⁺ cells between EV (+) and EV (-). n.s., no significance.

- Similarly, the functional experiments presented in figure 6 are essential and should be complemented to fully prove the role of EVs. First, the phenotypes should be more detailed: why are there more GFP⁺ cells after treatment? Increased proliferation or decreased cell death? Other phenotypes, such as cell migration, cell fusion (by measuring the proportion of multinucleated cells for instance) and differentiation should be quantified. Finally, EVs isolated from in vitro cultured OBs as mentioned in the first point should be tested on OCs.

The reviewer has pointed out an important issue that has been addressed in our revised manuscript. In order to clarify why the number of *trap:GFP*⁺ OCs increases by treatment with OB-derived EVs, kidney marrow cells (KMCs, hematopoietic cells) cultured with or without OB-derived EVs were further characterized. After plating KMCs (approx. 60,000 cells) from *trap:GFP* animals on a fibronectin-coated plate, KMCs were treated with or without mCh⁺ Hoe^{low} EVs (2,000 particles) from fractured scales. At 2 days post-culture, we counted the number of GFP⁺ cells by fluorescent microscopy, followed by counting the total number of hematopoietic cells harvested from the plate (both GFP⁺ and GFP⁻ cells) using a hemocytometer. We found that although the number of GFP⁺ cells increased by treatment of EVs, the total number of hematopoietic cells was unchanged (new Supplementary Fig 4a, b). These results suggest that treatment of EVs does not affect hematopoietic cell proliferation, but does affect OC differentiation from hematopoietic cells.

We next examined the percentage of multi-nucleated GFP⁺ cells by staining with Hoechst 33342 at 2 days post-culture. As shown in a new Supplementary Fig 4c, the percentage of multinucleated GFP⁺ cells significantly increased by treatment with EVs, whereas of the percentage of mononucleated GFP⁺ cells decreased. We also observed a few GFP⁺ cells with more than three nuclei in KMCs treated

with EVs. These results suggest that treatment with EVs also promotes cell fusion of OCs.

Finally, we examined the motility of *trap:GFP*⁺ cells by confocal microscopy. As shown in new Fig 2e, *trap:GFP*^{high} OCs isolated directly from the fractured scale were classified into three types based on their morphology and number of nuclei. We found that type-2 OCs, which have many protrusions with one or two nuclei, actively migrated in vitro. In contrast, type-1 OCs, which are round with a single nucleus, and type-3 OCs, which have more than three nuclei, exhibited poor motility (new Supplementary Movie 3 and 4). We also examined the motility of *trap:GFP*⁺ OCs induced from KMCs in vitro. Due to the high density of cells in the well, however, the majority of both EV-treated and untreated GFP⁺ cells migrated very slowly, or did not move (data not shown), while some GFP⁺ cells showed the morphological features of type-2 OCs. Unfortunately, this cell culture assay may not be suitable to evaluate the effect of EV treatment on the motility of OCs. Taken together, our data suggest that treatment of OB-derived EVs promotes at least differentiation and fusion of OCs from hematopoietic cells in vitro.

As described above, it is currently difficult to culture OBs in zebrafish. Therefore, we cannot show the role of EVs from cultured OBs in the present study.

These new data have been added to new Supplementary Fig 4. In addition, the movies of type-1, -2, and -3 OCs have been added as new Supplementary Movie 3 and 4.

- Another important statement of the paper is the involvement of Rank signalling in the differentiation of OCs. Here the authors show the expression of Rank and Rankl transcripts in EVs isolated by flow cytometry and use a CRISPR-induced KO of Rank in the whole fish to test its role in OCs differentiation. Alternative approaches should be used to 1) really prove the presence of Rankl protein on OBs EVs surface and 2) properly demonstrate the function of EV Rankl in OCs differentiation. Could the authors test if any available anti-mammalian Rankl antibody works on zebrafish ? In addition, it should be determined whether rankl KO affects EV secretion.

We have tested two different antibodies against human RANKL (sc-377079) and mouse Rankl (sc-52950). Unfortunately, these antibodies did not work in zebrafish based on immunofluorescence analysis (data not shown). This is probably due to the low conservation of amino acid sequences in Rankl proteins between mammals and zebrafish (the identity was approximately 25%) (Response Table). To date, there is no antibody against zebrafish Rankl. Therefore, it is currently difficult to confirm the expression of Rankl proteins on OB-derived EVs.

Response Table. % Identity of Rankl proteins

D. rerio vs. H. sapiens	25.5
D. rerio vs. M. musculus	24.7
H. sapiens vs. M. musculus	85.1

According to the reviewer's suggestion, we compared the absolute number of OB-derived EVs in a fractured scale between wild type and *rankl* gRNA animals. We found that the number of OB-derived EVs was unchanged in the fractured scale of *rankl* gRNA animals compared with wild type animals (new Supplementary Fig. 5e). Since the number of OBs was also unchanged in *rankl* gRNA animals (Supplementary Fig. 5e), our data suggest that knockdown of *rankl* does not affect secretion of EVs in OBs.

These new data were added to Supplementary Fig. 5e, and the sentences in results have been modified to reflect these new data.

Other points:

- Some phenotypes should be better described. For instance, the proportion of the different types of OBs and OCs observed in culture (figure 2) should be quantified, before and after scale fracture.

According to the reviewer's suggestion, we have classified OB and OC subsets into three types and quantified the frequency of each type in the fractured scale. Because of the low frequency of GFP^{high} cells in intact scales, it is difficult to compare the frequency of OC subtypes between intact and fractured scales.

These new data were added to Fig 2c-e, and sentences in results were modified to reflect these new data.

The in vivo movies would also benefit from some quantifications: for instance, what is the concentration of EVs with and without scale fracture ?

We believe that flow cytometric analysis is more reliable to quantify the number of EVs than in vivo imaging analysis. Therefore, we compared the absolute number of OB-derived EVs between an intact and fractured scale by flow cytometry. As shown in new Fig. 5e, the number of OB-derived EVs was approximately three times higher in the fractured scale than that in the intact scale.

This new data was added to Fig. 5e, and the sentences in results were modified to reflect these new data.

- Some experimental details are lacking. For instance, in the very nice movie, dissected in figure 4, what type of imaging are we looking at ? If this is confocal, is it single planes or Z-projections ? If this is not confocal, how can the authors make sure that the particles have been internalized ? Another example is the functional experiments presented in figure 6: how are the experiment normalized ? are similar numbers of cells/EVs added to OCs, or similar amounts of proteins ? This should be specified.

We apologized for incomplete explanations of our experimental methods. We used confocal microscopy for all in vivo live-imaging analyses and showed most images as Z-projection, including Fig. 4 and Supplementary Movie 6 (old Supplementary Movie 4). To validate the internalization of an mCherry⁺ particle in the GFP⁺ cell, single plane images have also been added to new Fig. 4. We confirmed that the GFP⁺ cell actually possessed an mCherry⁺ particle in its cytoplasm. The new images have been added to new Fig. 4b and we have modified the sentences in materials and methods and results to better explain these experiments.

In the cell culture experiment shown in new Fig. 7c, d, and Supplementary Fig. 4, experiments were performed with addition of the same number of sorted OBs or EVs (2,000 cells or particles) to the same number of KMCs (60,000 cells).

- Description of the compartments storing OBs particles within OCs cells is an important aspect. Authors could costain for late endosomes/lysosomes, for instance using lysotracker probes. It could allow to identify differences in the uptake of MVs or Abs (figure 6).

Although the reviewer suggested to use lysotracker probes, we did not find evidence that lysotracker probes can separate internalized MVs and ABs within the cell. Lysotracker is a fluorescent probe that labels acidic organelles, such as late endosome/lysosomes. There are some reports showing colocalization of lysotracker probes with fluorescently labeled exosomes, MVs, and ABs (Yuyama et al., J Biol Chem. 2012; Dutta et al., PLoS One. 2014; Saari et al., J Control Release. 2015; Li et al., J Neuroinflammation. 2014; Wager et al., PLoS One. 2016), suggesting that any types of EVs are distributed in late endosome/lysosomes at some timing. As far as we know, the internalization route of MVs and ABs in the cell is still not fully understood, and it is unclear if MVs and ABs can be distinguished by staining of lysosomes. Unfortunately, therefore, it is not currently feasible to distinguish internalized MVs and ABs in OCs.

- In figure 2F, authors show an EM picture with arrows pointing to “lysosomes” ; these compartments are clearly a mixture of different types, including early endosomes and multi-vesicular bodies. More

convincing images should be presented or the legends should be modified.

Thank you very much for pointing out this issue. Since we also considered that the arrowheads in Fig. 2f point to multiple types of vesicles, we described these vesicles as “lysosome-like vesicles”. For clarity, we now describe these vesicles as, “various types of vesicles including secondary lysosomes, early endosomes, and multi-vesicular bodies” and modified the results and figure legend to reflect this change.

- In the discussion, the section dedicated to the study of EVs in vivo should be more detailed, as this paper presents valuable novel model and data to the field. In particular, recent papers describing the use zebrafish in EVs tracking should be cited (reviewed in Verweij et al. Trends in Cell Biology 2019).

Thank you very much for this suggestion. We have added the new reference suggested by the reviewer to the discussion and detailed the advantages of zebrafish in the study of EVs in the revised version of the manuscript.

REVIEWERS' COMMENTS:

Reviewer #1 (Remarks to the Author):

The authors have responded adequately to most of the points made, but I have two suggestions with amendments to writing that relate to the emphasis of this paper and its biological appeal.

The emphasis of this paper on scale biology has still not been made sufficiently clear. The Introduction begins with several lines about bone and its diseases. A more appropriate way of beginning would be to emphasize the fact that the work is on cell biology in fish scales. The point is made clearly in paragraph 3 of Introduction. The first three lines of paragraph 1 are not an appropriate starting point for the introduction. This work is not about bone biology. It uses novel methods to produce interesting data in fish scale biology that is most likely relevant to the processes of osteoblast-osteoclast communication in mammalian biology. It certainly confirms in the fish scale what has been already found in mammalian cells concerning osteoblast-derived vesicles promoting osteoclast formation

With regard to Point No.2 made in my review, more needs to be done than simply refer in a sentence to refs 14 and 15. They are clearly very directly important for the present work, and should be discussed properly in the Discussion section. These authors have developed a most valuable double-labelled transgenic zebra fish and have used the scale fracture model in excellent experiments to produce interesting data. It complements and confirms the mammalian data, and shows very clearly that the vesicles are engulfed by osteoclasts in vivo. It is much more important that this be acknowledged and incorporated adequately into Discussion by putting the work into context with what has already been shown in mammalian cells. This could profitably replace the several lines (299-304) spent discussing the irrelevant findings that there are "static resorbing" and "moving non-resorbing" osteoclasts.

Reviewer #2 (Remarks to the Author):

I appreciate the revisions performed by the authors and believe that the manuscript has been improved and can be published in its present form.

Remaining comment:

- The image in figure 5C is puzzling. It shows a pretty large structure (2x3 um), containing internal compartments (mitochondria, endosomes ..). Is this image really representative of the mCh+ Hoelw EVs ? It looks rather big compared to the size distribution presented in figure 5E. The authors should find a more representative image or explain their choice.

Responses to Reviewers (Kobayashi-Sun et al., *Communications Biology*)

We thank the reviewers for their helpful suggestions again. We believe the manuscript is properly improved in response to reviewer comments.

Reviewer #1

The emphasis of this paper on scale biology has still not been made sufficiently clear. The Introduction begins with several lines about bone and its diseases. A more appropriate way of beginning would be to emphasize the fact that the work is on cell biology in fish scales. The point is made clearly in paragraph 3 of Introduction. The first three lines of paragraph 1 are not an appropriate starting point for the introduction. This work is not about bone biology. It uses novel methods to produce interesting data in fish scale biology that is most likely relevant to the processes of osteoblast-osteoclast communication in mammalian biology. It certainly confirms in the fish scale what has been already found in mammalian cells concerning osteoblast-derived vesicles promoting osteoclast formation

Thank you very much for pointing out the remaining problems in the introduction. As the reviewer mentioned, we have omitted the first three lines regarding bone and its diseases, and the paragraph 3 of the previous version of the manuscript has been placed in the first paragraph of the revised version to emphasize that the zebrafish scale model is useful for the study of OC - OB communication.

With regard to Point No.2 made in my review, more needs to be done than simply refer in a sentence to refs 14 and 15. They are clearly very directly important for the present work, and should be discussed properly in the Discussion section. These authors have developed a most valuable double-labelled transgenic zebra fish and have used the scale fracture model in excellent experiments to produce interesting data. It complements and confirms the mammalian data, and shows very clearly that the vesicles are engulfed by osteoclasts in vivo. It is much more important that this be acknowledged and incorporated adequately into Discussion by putting the work into context with what has already been shown in mammalian cells. This could profitably replace the several lines (299-304) spentg discussing the irrelevant findings that there are “static resorbing” and “moving non-resorbing” osteoclasts.

According to the reviewer's suggestion, we have added some sentences in the last paragraph of Discussion regarding Huynh et al., 2016 and Ikebuchi et al., 2018 (new ref. 26 and 27, respectively) as followed:

“In mammals, EVs derived from OCs also play an important role in osteoclastogenesis as well as osteoblastogenesis. EVs from OC precursors promote OC formation in whole bone marrow cultures, whereas EVs from OC-enriched cultures inhibit osteoclastogenesis²⁶. Moreover, EVs released by mature OCs contain a high level of RANK and promote bone formation by triggering RANKL reverse

signaling in OBs²⁷. These observations strongly suggest that cell-cell communication in osteoclastogenesis and osteoblastogenesis is largely dependent on the release and uptake of EVs. Such EV-mediated intercellular communication represents a novel cellular mechanism in the bone. Further analysis of EVs in bone tissue will elucidate the molecular mechanisms that regulate the balance between bone resorption and formation.”

Ishii's group (ref. 43, 44) found using a unique imaging system in mice that there are two subsets of functional OCs in terms of their motility and function. This observation is closely associated with one of our findings that *trap:GFP^{high}* cells in the fractured scale contain different types of OCs in terms of motility and morphology. Although it is still unclear if these different types of OCs in mice and zebrafish are equivalent or not, we believe these descriptions in our manuscript are important for the study of bone tissue.

Reviewer #2

- The image in figure 5C is puzzling. It shows a pretty large structure (2x3 um), containing internal compartments (mitochondria, endosomes ..). Is this image really representative of the mCh⁺ Hoel^{ow} EVs ? It looks rather big compared to the size distribution presented in figure 5E. The authors should find a more representative image or explain their choice.

We appreciate this suggestion. To display the structure of EVs in a better resolution, we selected a large EV, which contains cell compartments including mitochondria and vesicles. As the reviewer mentioned, however, these large EVs are minor in the mCh⁺ Hoel^{ow} fraction (Fig. 5c). We have therefore added a small EV to Fig. 5b as well. The sentences in Results and figure legends have been slightly modified to reflect these changes.